# Correlation of two different devices for the evaluation of primary implant stability depending on dental implant length and bone density: An in vitro study

Jungwon Lee[1], Young-Jun Lim[2], Jin-Soo Ahn[1], Bongju Kim[3], Yeon-Wha Baek[4]*, Bum-Soon Lim[1]*

1 Department of Dental Biomaterials Science, School of Dentistry, Dental Research Institute, Seoul National University, Seoul, 03080, Republic of Korea, 2 Department of Prosthodontics and Dental Research Institute, School of Dentistry, Seoul National University, Seoul, Republic of Korea, 3 Dental Life Science Research Institute, Seoul National University Dental Hospital, Seoul, Republic of Korea, 4 Department of Prosthodontics, Gwanak Center, Seoul National University Dental Hospital, Seoul, Republic of Korea

* obero7@snu.ac.kr (Y-WB); nowick@snu.ac.kr (B-SL)

**Data Availability Statement:** All relevant data are within the article and its Supporting information files.

## Abstract

Non-invasive objective implant stability measurements are needed to determine the appropriate timing of prosthetic fitting after implant placement. We compared the early implant stability results obtained using resonance frequency analysis (RFA) and damping capacity analysis (DCA) depending on the implant length and bone density. Total 60, 4.0 mm diameter implants of various lengths (7.3 mm, 10 mm, and 13 mm) were used. In Group I, low-density bone was described using 15 PCF (0.24 g/cm3) polyurethane bone blocks, and in Group II, 30 PCF (0.48 g/cm3) polyurethane bone blocks were used to describe medium density bone. RFA was performed using an Osstell® Beacon+; DCA was performed using Anycheck®. Measurements were repeated five times for each implant. Statistical significance was set at P <0.05. In Group I, bone density and primary implant stability were positively correlated, while implant length and primary implant stability were positively correlated. In Group II, the implant stability quotient (ISQ) and implant stability test (IST) values in did not change significantly above a certain length. Primary implant stability was positively correlated with bone density and improved with increasing implant length at low bone densities. Compared with the Osstell® Beacon+, the simplicity of Anycheck® was easy to use and accessible.

## Introduction

The healing period after dental implant placement can vary significantly from patient to patient and is influenced by various factors such as systemic health, bone quality, and periodontal status. During this healing period, the implant undergoes a process called osseointegration, where it fuses with the surrounding bone, providing a stable foundation for the prosthetic tooth or crown. To ensure successful osseointegration and reduce the risk of

**Funding:** This work was supported by the Korea Medical Device Development Fund grant funded by the Korea government (the Ministry of Science and ICT, the Ministry of Trade, Industry and Energy, the Ministry of Health & Welfare, the Ministry of Food and Drug Safety) (Project Number:1711138936, RS-2020-KD000291). The funders had no role in study design, data collection and analysis, decision to publish, or preparation of the manuscript.

**Competing interests:** NO authors have competing interests.

complications, it is crucial to monitor the implant's stability throughout the healing phase. Early loading of implants that have not adequately integrated with the bone can lead to implant failure, while waiting too long to load the implant can extend the treatment time unnecessarily.

Non-invasive implant stability evaluation methods include radiographic evaluation, percussion tests, and insertion torque. While radiographic evaluation is non-invasive and provides valuable information about the condition of the implant and surrounding bone, there are certain issues to consider. Radiographic images can suffer from distortion, which may affect the accuracy of the assessment [1]. Additionally, relying solely on changes in radiographic bone levels might not be sufficient to accurately predict implant stability, especially since bone remodeling and changes in bone levels can be influenced by various factors beyond just implant stability [2]. Percussion tests involve tapping or lightly striking the implant or prosthetic tooth to assess its stability based on the sound or vibration produced. However, these tests are subjective and heavily dependent on the experience and skill of the dentist. As a result, the results may not always be consistent or reliable. This method involves measuring the rotational force required to insert the implant into the bone, which can provide information about bone quality and initial stability. While insertion torque is a relatively objective indicator, it has its limitations. For instance, it may not accurately reflect the long-term stability of the implant, and it does not assess lateral or longitudinal mobility, which are important aspects of stability.

An objective method to measure implant stability is resonance frequency analysis (RFA), which evaluates the stability of implants using sinusoidal signals and small transducers, as introduced by Meredith et al. [3]. Representative measuring equipment included the Osstell ISQ and Osstell® Beacon. Osstell is used to tighten a magnetic SmartPeg coated with zinc on an ingrained implant. Using of a turning fork, magnetism is sent to the SmartPeg to obtain resonant vibration and osseointegration between the implant and alveolar bone is measured indirectly. The Osstell device records the resonant frequency of the implant-bone interface, which is known as the Implant Stability Quotient (ISQ). The ISQ value is represented on a scale from 1 to 100. ISQ values can be used to assess implant stability. Typically, an ISQ value of less than 60 indicates low stability, between 60 and 69 suggests moderate stability, and a value of 70 or higher indicates high stability. In general, the higher the ISQ value, the greater the implant stability.

While RFA is non-invasive and provides valuable information about the implant's stability, it does have certain drawbacks that should be considered. RFA provides an indirect measurement of implant stability based on resonant vibration. While it offers valuable data about osseointegration, it does not directly measure longitudinal or lateral perturbations or the true mechanical stability of the implant. To perform RFA, a separate instrument, the SmartPeg, is required to be attached to the implant fixture. This may involve removing the healing abutment, which can cause inconvenience and a risk of potential complications during the healing phase. The torque applied when tightening the SmartPeg can affect the reliability of the resulting ISQ value [4,5]. There is a lack of consensus regarding the optimal torque value for tightening the SmartPeg, which may introduce variability in measurements. The hand tightening of the SmartPeg may introduce subjectivity and variability as different operators may apply different levels of finger pressure, leading to inconsistent results.

The Damping Capacity Analysis (DCA) is another objective method used to measure implant stability. In this technique, a certain amount of force is mechanically applied to the implant post and the fluctuation of the implant in both the longitudinal and lateral directions is measured. A typical measurement device is the PerioTest M. Measured values from the PerioTest are affected by the angle of impact and high strength of the blow, and the number of

blows is high (16 times) causing a feeling of rejection in the patient. In addition, the PerioTest has low reliability [6]. The recently developed modified damping capacity analysis device (Anycheck®, Neobiotech Co., Ltd., Seoul, Korea) is highly reproducible and can be measured by direct contact with the object by improving the striking method [7]. This device evaluates the osseointegration between the implant and alveolar bone by measuring the time the striking rod (head) comes into contact with the implant or abutment. The measurement obtained from the modified device is called the Implicit Stability Test (IST) value, expressed as a number ranging from 1 to 99. The color-coding of IST values helps to classify implant stability. IST values in red range from 1 to 59, indicating lower stability. IST values in orange range from 60 to 64, suggesting moderate stability. and IST values in green range from 65 to 99, indicating higher stability. One of the challenges in implant placement is the quality of alveolar bone and critical anatomical structures. The quality of D4 bone density is generally described as poor because it is soft, and it is difficult to obtain primary stability from implants [8]. Low-density bone implant sites have been reported as the greatest potential risk factor for implant loss when working with standard bone-drilling protocols [9]. The bone density at the implant site plays a critical role in determining the initial stability of the implant. Dense cortical bone (D1 and D2) provides better initial stability due to its higher density and resistance to implant movement. On the other hand, lower density bone (D4) may result in decreased primary stability due to its lower resistance to implant movement [10]. For this reason, research is ongoing to compensate for poor primary stability at low bone density. Primary implant stability increases with a larger implant diameter because the contact area between the implant and the bone increases [11]. In this study, we aimed to compare the changes in primary implant stability by varying the length rather than the diameter of the implant.

This study primarily aimed to compare the values obtained by expert and non-expert using two different devices for primary stability according to dental implant length and artificial bone density and to investigate the correlation of results from the two devices.

## Materials and methods

### Preparation of artificial bone blocks and dental implants

In this study, 4.0 mm diameter internal connection type implants (IS-III Active, Neobiotech, Seoul, Korea) of various lengths (7.3 mm, 10 mm, and 13 mm) were used (Fig 1). Polyurethane bone models (Sawbones; Pacific Research Laboratories Inc., Washington, DC, USA) were used to simulate cancellous bone, and the size of the artificial bone block was 130 mm × 90 mm × 40 mm. Two different types of polyurethane bone models were compared: one with a uniform density of 15 PCF (0.24 g/cm3, Group I) and the other with a uniform density of 30 PCF (0.48 g/cm3, Group II).

### Surgical procedure and implant placement

Sixty implants were used, 30 in each group and they consisted of three different lengths (10 implants each): 7.3 mm, 10 mm, and 13 mm. All implants used in the study were inserted at a constant depth and vertical angle to the bone blocks using a specially designed implant placement & drilling machine (Hangil Technics, Gyeonggi, Korea) (Fig 2) with self-tapping for standardization.

The implant placement site was prepared using two drilling protocols according to the manufacturer's instructions [12]. In Group I, a ∅ 2.2 mm initial drill and ∅ 3.0 mm taper drills were used, and in Group II, a ∅ 2.2 mm initial drill, final drills of ∅ 3.0 mm and ∅ 3.5 mm, and a ∅ 3.5 mm tap drill were used. Both groups used the Neo Master Kit (Neobiotech, Seoul,

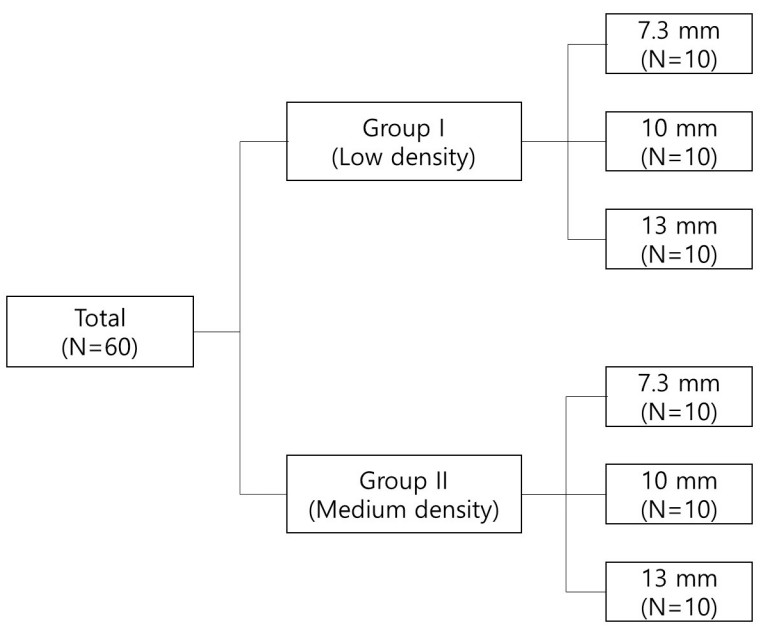

**Fig 1. The schematic diagram.**

Korea) and were drilled at 1,200 rpm. The insertion torque was set around 18 Ncm for group I and 35–40 Ncm for group II. Each implant was placed 30 mm apart.

## Implant stability measurements

The observer consisted of expert and non-expert. An expert refers to a dental hygienist with five years of experience proficient in using measuring device, while a non-expert denotes someone who has never used measuring device. The RFA measurements were performed using an Osstell® Beacon+ (Integration Diagnostics, Göteborg, Sweden). Before the implant stability measurements were made, the bone block was firmly fixed to the vise. A type 5 Smart-Peg was fastened to the implant using a plastic mount at 4–6 Ncm by hand tightening, according to the manufacturer's instructions. Measurements were performed in four directions (three times in each direction) at a distance of 3–5 mm and at an angle of 45˚, and the ISQ measurements were averaged in the four directions for each implant (Fig 3). This procedure was repeated five times [13].

The DCA was performed using Anycheck® (Neobiotech, Seoul, Korea). For the measurements, a Ø4.8 × 4 mm healing abutment was tightened with a constant force of 10 Ncm using a torque ratchet and torque wrench. A 10˚ jig was made using a polyvinyl siloxane impression material (putty) to maintain a constant upward angle of 0˚ to 30˚ with respect to the ground following the manufacturer's manual. Five replicates were recorded as the average IST values measured in two implant directions (Fig 3).

## Statistical analysis

A paired t-test was performed to verify whether the ISQ and IST values of the two bone densities (Group I and Group II) demonstrated a significant difference. Simple linear regression analysis was also applied to assess the effect of bone density (15 PCF and 30 PCF) on ISQ and IST values. One-way ANOVA was performed to verify that the ISQ and IST values of the

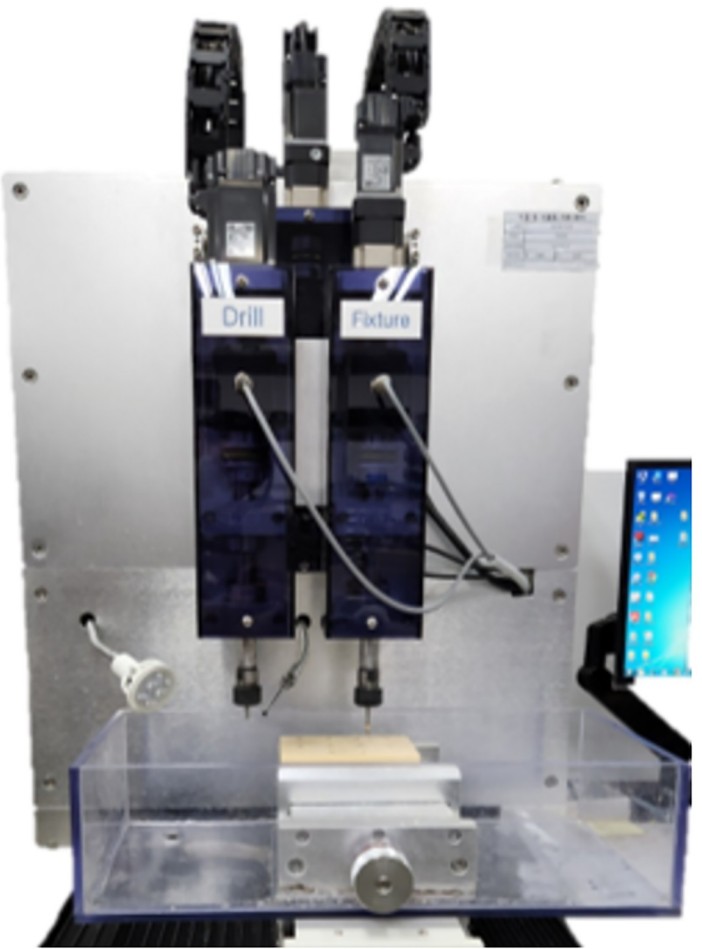

**Fig 2. Specially designed implant placement & drilling machine.**

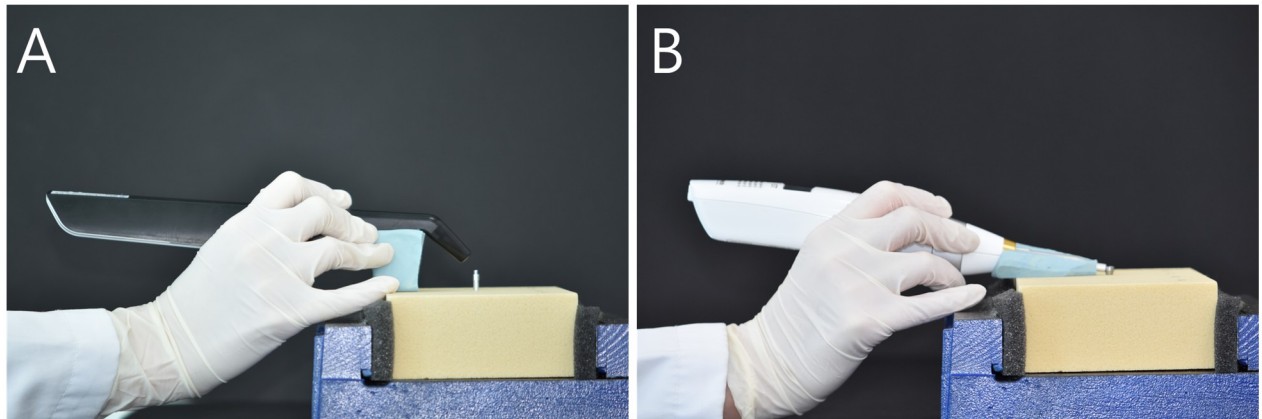

**Fig 3. Primary implant stability measurement.** (A) Osstell® Beacon+, (B) Anycheck®.

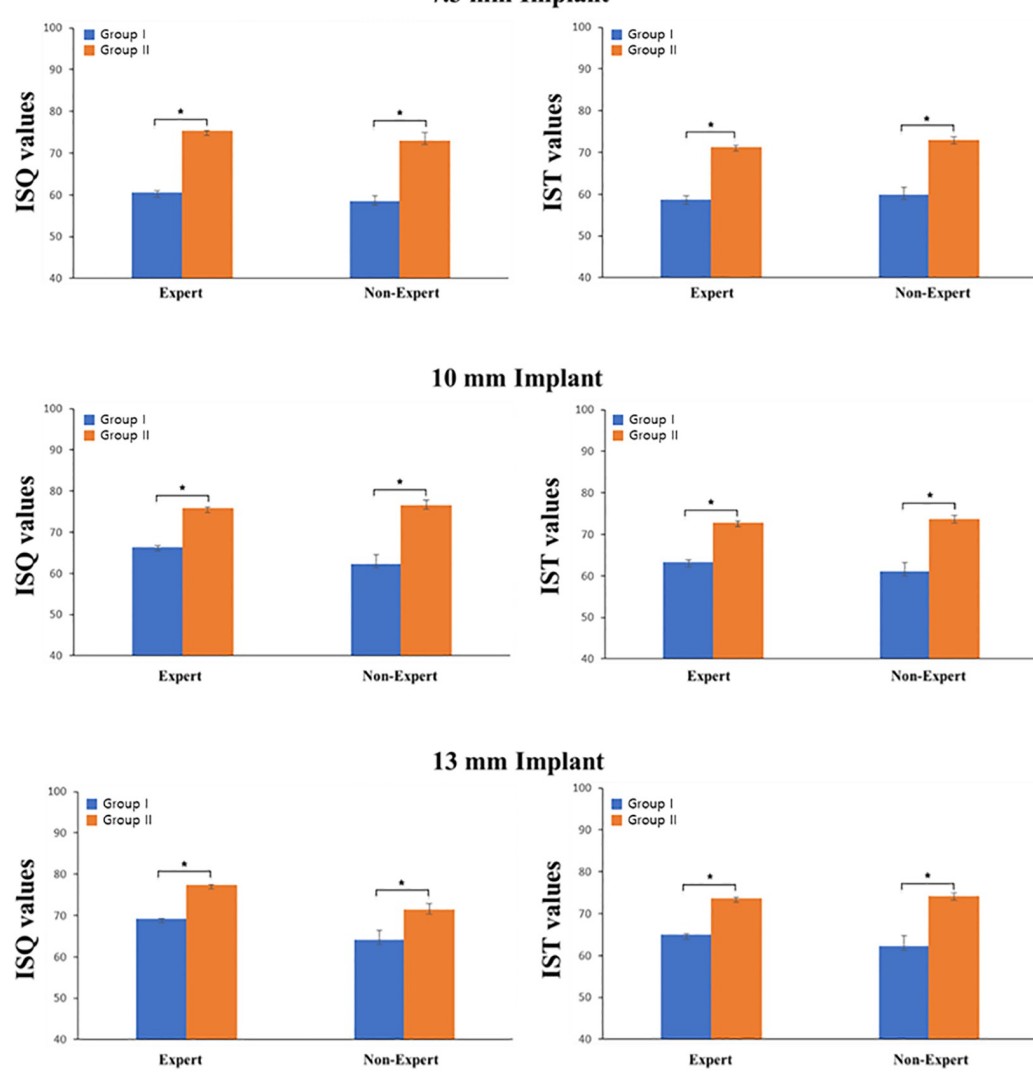

**Fig 4. The implant stability quotient and the implant stability test value depending on density.** * $P < 0.05$.

three-implant length (7.3 mm, 10 mm, and 13 mm) reported a significant difference. Scheffe and Duncan were used for post hoc. Simple linear regression analysis was also applied to assess the effect of the implant length (7.3 mm, 10 mm, and 13 mm) on the ISQ and IST values. Simple linear regression analysis was used to confirm the correlation between the ISQ and IST values. All calculations were conducted using SPSS software (version 25, SPSS), and significance was defined as $P < 0.05$.

## Results

### Effect of bone density

The difference in primary stability depending on bone density is illustrated in Fig 4. The difference in ISQ values according to bone density was as follows: In the artificial bone block with

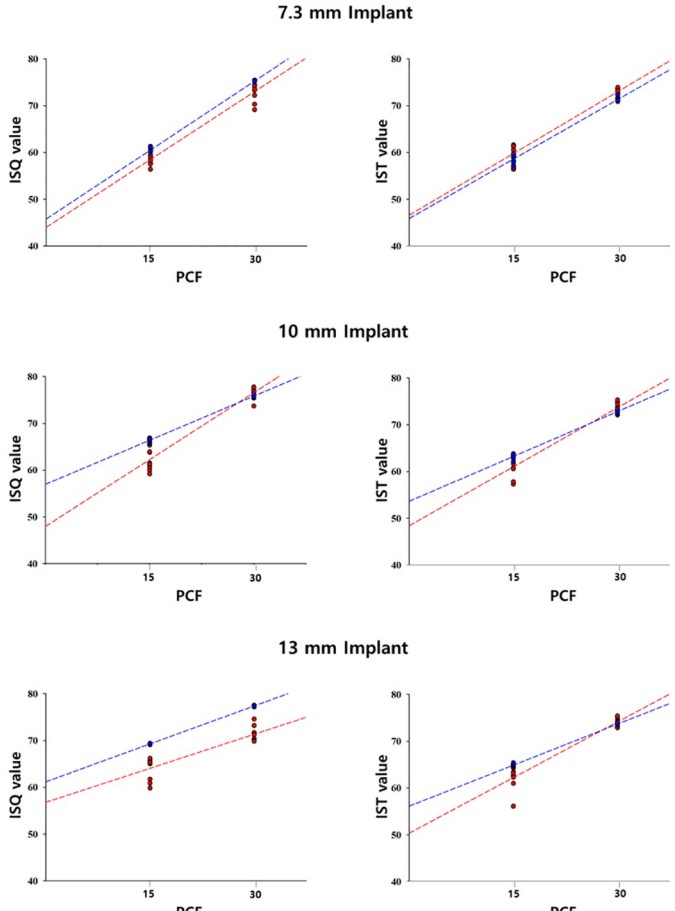

**Fig 5. Correlation between the implant stability quotient and implant stability test value versus bone density.**

7.3 mm implants, in Group I, the mean for the expert was 60.52 and 58.51 for the non-expert. In Group II, with medium density, the mean for the expert was 75.28, and 73.02 for the non-expert, with a significant difference between the two groups ($P$ <0.001). At 7.3 mm, the correlation coefficient (R) between bone mineral density (BMD) and the ISQ value was 0.999 and 0.978 for the expert and the non-expert, respectively. In the artificial bone block with 10 mm implants, in Group I, the mean for the expert 66.40, and 62.27 for the non-expert. In Group II, the mean for the expert was 75.82, and 76.57 for the non-expert, with a significant difference between the two groups ($P$ <0.001). The correlation coefficient(R) between BMD and the ISQ value at 10 mm was 0.998 and 0.972 for the expert and the non-expert, respectively. In the artificial bone block with 13 mm implants, in Group I, the mean for the expert was 69.30, and 64.08 for the non-expert. In Group II, the mean for the expert was 77.40, and 71.36 for the non-expert, with a significant difference between the two groups ($P$ <0.001). The correlation coefficient(R) between BMD and the ISQ value at 13 mm was 1.000 and 0.891 for the expert and the non-expert, respectively.

The differences in IST value according to bone density were follows: In the artificial bone block with 7.3 mm implants, in Group I, the mean for the expert was 58.61, and 59.80 for the non-expert. In Group II, with medium density, the mean for the expert was 71.33, and

73.00 for the non-expert, with a significant difference between the two groups ($P <0.001$). The correlation coefficient (R) between BMD and the IST value at 7.3 mm was 0.994 and 0.980 for the expert and the non-expert, respectively. In the artificial bone block with 10 mm implants, in Group I, the mean for the expert was 63.24, and 61.03 for the non-expert. In Group II, with medium density, the mean for the expert was 72.83, and 73.64 for the non-expert, with a significant difference between the two groups ($P <0.001$). The correlation coefficient (R) between BMD and the IST value at 10 mm was 0.995 and 0.971 for the expert and the non-expert, respectively. In the artificial bone block with 13 mm implants, in Group I, the mean for the expert was 64.91, and 62.25 for the non-expert. In Group II, with medium density, the mean for the expert was 73.68, and 74.14 for the non-expert, with a significant difference between the two groups ($P <0.001$). The correlation coefficient (R) between BMD and the IST value at 13 mm was 0.999 and 0.962 for the expert and the non-expert, respectively.

As presented in Fig 5 and Table 1, bone density is highly correlated with primary implant stability at all lengths. The regression coefficient significance test revealed a significant positive correlation between bone density and primary implant stability. Therefore, the higher the bone density, the higher the primary implant stability.

## Effect of implant length

The difference in ISQ value according to the implant length is presented in Fig 6. In Group I, (low-density), For the expert, there was a significant difference among all lengths ($P < 0.0001$). For the non-expert, there was a significant difference between 7.3 mm and 10 mm, and between 7.3 mm and 13 mm ($P = 0.008$, $P < 0.0001$). There was no statistically significant difference between 10 mm and 13 mm; however, the ISQ value increased with length. In Group II, which had medium density, there was a significant difference among all lengths for the expert ($P < 0.05$). For the non-expert, there was a significant difference between 7.3 mm and 10 mm, and between 10 mm and 13 mm ($P < 0.0001$).

The differences in IST value according to implant length are presented in Fig 6. In Group I, For the expert, there was a significant difference among all lengths ($P < 0.0001$). For the non-expert, there was a significant difference between 7.3 mm and 13 mm ($P < 0.05$), however unlike the other observers, the IST value decreased with increasing length. There was not statistically significant difference between 7.3 mm and 10 mm, however the IST value increased with increasing length. There was no statistically significant difference between 10 mm and 13 mm, however the IST value decreased with increasing length. In Group II, For the expert, there was a significant difference among all lengths ($P < 0.0001$). For the non-expert, there was a significant difference between 7.3 mm and 13 mm ($P = 0.011$). There were no statistically significant differences between the other lengths; however, the IST value increased with increasing length.

**Table 1. Primary stability depending on density.** R: Correlation Coefficient; ISQ: Implant Stability Quotient; IST: Implant Stability Test.

| | | | 7.3 mm | 10 mm | 13 mm |
|---|---|---|---|---|---|
| $R^2$ | ISQ value | Expert II | 0.998 | 0.996 | 0.999 |
| | | Non-Expert | 0.956 | 0.944 | 0.793 |
| | IST value | Expert II | 0.988 | 0.990 | 0.998 |
| | | Non-Expert | 0.961 | 0.944 | 0.925 |

### Group 1(Low density)

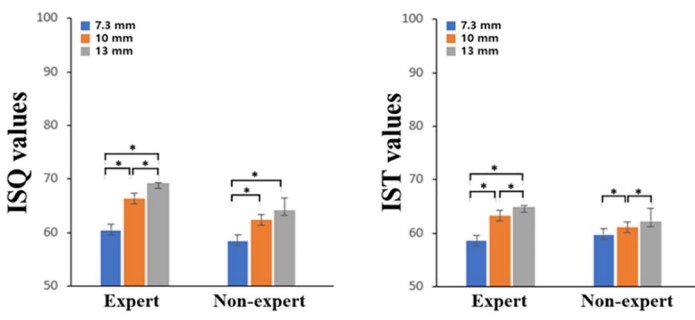

### Group 2(Medium density)

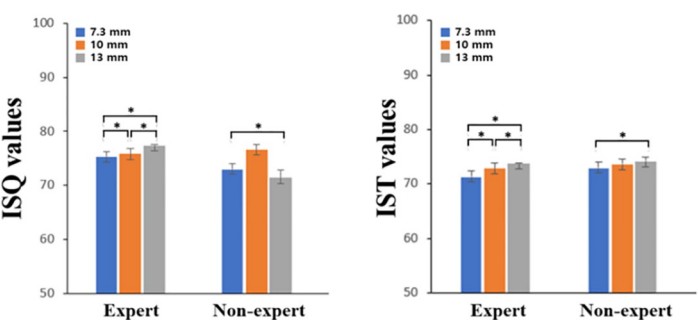

**Fig 6. The implant stability quotient and the implant stability test value depending on fixture length.** * $P < 0.05$.

## Correlation between the implant stability quotient and the implant stability test value versus density and fixture length

The changes in the ISQ and the IST value with density are presented in Fig 5. For both the Osstell® Beacon+ and Anycheck®, primary implant stability increased with increasing bone density, regardless of implant length.

In Fig 7, in Group II, the primary stability did not display any specific change with increasing implant length; however, in Group I, the primary implant stability increased with increasing implant length (Table 2).

## Discussion

In line with the trend towards continuous monitoring using objective and qualitative methods to determine the status of implant stability, this study analyzed the values of measurement devices using RFA and DCA, which are commonly used to measure implant stability in clinical practice. Moreover, an experiment was designed to investigate the trends in the ISQ and IST values with changes in bone density and implant length.

Previous studies comparing different implant stability measurement devices were performed using pig bones [14,15]. The use of this particular biological sample can result in variability in bone quality owing to factors such as different bone densities, depending on the distribution of heterogeneous bone cells in the cross-section or the site of the specimen [16]. Artificial bones were used to eliminate the confounding variables. Although an artificial bone

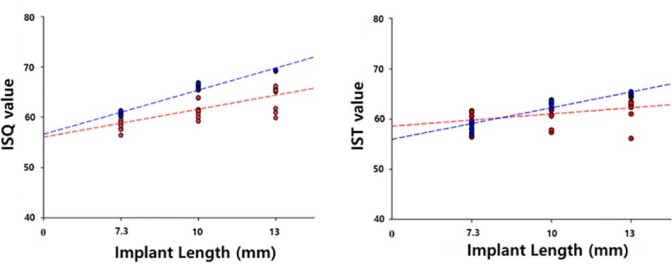

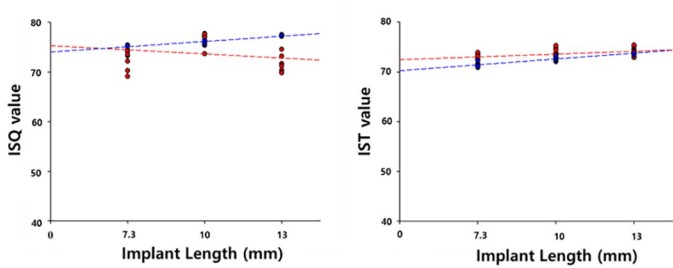

**Fig 7. Correlation between the implant stability quotient and implant stability test value versus fixture length.**

**Table 2. The implant stability quotient (ISQ) and the implant stability test (IST) value depending on fixture length.**

| Group | | ISQ value | | IST value | |
|---|---|---|---|---|---|
| | | Expert | Non-Expert | Expert | Non-Expert |
| I (Low density) | $R^2$ | 0.956 | 0.573 | 0.885 | 0.197 |
| II (Medium density) | $R^2$ | 0.900 | 0.066 | 0.858 | 0.264 |

cannot fully mimic the viscoelastic properties of actual bone tissue, it has the advantage of having the density, size, and shape of bone to be consistent and can be modeled in the most necessary forms. These advantages allowed us to represent the structure of the human cancellous bone as closely as possible [17].

Previous studies [18–20], have suggested that a distinct layer of cortical bone on marginal bone plays a decisive role in the clinical value of RFA, whereas trabecular bone has a minor influence on the implant stability compared with the marginal bone density. The cortical bone layer in biological samples, as well as the structure of the bone, affects implant stability [18]. Kanthanat Chatvaratthana et al. found that the ISQ value was highly correlated with cortical bone thickness [19]. Previous studies have used models with a cortical bone layer, so the results are not purely a function of variables such as the diameter or length of the implant. In fact, some studies have shown similar implant stability results when the bone density is different, but the cortical bone layer is the same thickness [20].

In this study, the experimental protocol was designed to completely exclude the influence of cortical bone and evaluate only the influence of implant length by using a uniform bone block without a cortical bone layer. By doing so, we sought to compare differences in implant stability due to pure variables of low and medium bone density, excluding other influences. To

compare the primary stability of the implant, the cancellous bone blocks were 15 PCF (0.24 g/cm3), which depicts low-density bone, and 30 PCF (0.48 g/cm3), which depicts medium-density bone.

Lekholm and Zarb reported high implant success rate in types 1–3 bone quality, whereas in a type 4 bone with little cortical bone layer, the success rate was low owing to the poor primary stability of the implant, resulting in no osseointegration [21]. Moreover, in a study conducted by Jaffin et al., the fixture failure rate was significantly higher in type 4 bones than in other types of bones [22]. These findings suggest that the bone quality is a major determinant of fixture loss. Hao et al. reported that the average bone density is the lowest in the maxilla, and the posterior maxilla is composed of D4 with a small cortical bone layer [23]. Based on the results of this study, we expect that it can be applied to maxillary premolars with little cortical bone layer and low bone density in clinical practice. In the artificial bone blocks without a cortical bone layer, the ISQ and IST values of Group 2 were significantly higher than those of Group I. By contrast, an attempt was made to implant the fixture in a bone block of 10 PCF (0.16 g/cm3). Unfortunately, proper implantation torque could not be achieved owing to poor bone quality, resulting in the elimination of the fixture and an inability to perform the experiment.

Baek et al. found that the ISQ values of patients with short implants were not significantly different from those of patients with regular implants, suggesting that the length of the implant did not affect its stability and prognosis [24]. Bischof et al. reported that primary implant stability demonstrated significant differences depending on the bone quality; however, implant diameter and length did not affect the primary implant stability [25].

However, unlike the above studies, there was a significant difference in primary implant stability according to implant length (Fig 4). In the low-density artificial bone block, there appeared to be a positive correlation between implant length and primary implant stability (Fig 5). However, in medium-density artificial bone blocks, there was either no difference or a decrease in the primary implant stability when implants with a length exceeding 10 mm were placed. Thus, at low densities, placing longer implants was effective in compensating for primary implant stability, whereas at medium density, longer implants were not necessarily beneficial to primary stability. Therefore, at a medium density, it is believed that placing a 10 mm implant is sufficient to achieve primary implant stability. This is because at high bone density, solid bone will hold the implant well regardless of the length of the implant [26].

In contrast, at low bone density, the longer the length of the implant, the greater the contact area of the bone has with the implant, which increases the stability of the implant. In addition, from a bio-mechanical perspective, many studies have reported that longer implants can lower the crown to implant (C/I) ratio and prevent alveolar bone loss and implant failure [27–29]. Therefore, when performing implant procedures on patients with poor bone quality, it can be expected that the primary stability of the implant will be complemented by the use of longer implants whenever possible.

Implant stability depends on the measurement device, angle, and observer [6]. The sensitivity and reliability of implant stability measurement devices are a topic of increasing interest. Buyukguclu et al. reported that experts with more than 4 years of experience measured primary implant stability with Osstell ISQ and Penguin RFA using RFA and found Osstell ISQ to be more reliable than Penguin RFA [30]. Lee et al. demonstrated the relative reliability of the Anycheck® device based on the reliability of the Periotest M using the percussive agitation method [11]. In this study, we conducted a pilot study to compare the effect of observer expertise on implant stability values and the usefulness of Osstell® Beacon+ and Anycheck® by analyzing the results of expert and non-expert rather than analyzing the reliability of the measurement equipment.

This is because when the Smartpegs were tightened to measure the ISQ value, the non-expert had difficulty applying a constant force and maintaining a constant distance using the contactless measurement method (Fig 3A). In contrast, Anycheck® is a contact measurement method (Fig 3B) and the measurement process is simple, so the difference between the IST values of the non-experts and experts is small. In this study, ISQ values for Osstell® Beacon + using RFA and IST values for Anyceck® using DCA displayed similar trends with changes in bone density and implant length, although the value was not consistent among observers (Figs 5 and 7). Therefore, it is crucial that implant stability measurements be performed by the same observer during follow-up appointments rather than relying on a specific measurement device.

This study has several limitations. Although we used artificial bone with the density specified by the International Organization for Standardization (ISO) 1183, we could not perfectly simulate the mechanical properties and clinical conditions of the actual in vivo bone. Furthermore, according to the manual, the most accurate IST value was obtained when the healing abutment and tare rod were perpendicular (90˚). Therefore, in this study, the jig was made at an angle as close to the vertical as possible to eliminate errors owing to the angular deviation during the measurement. However, in actual clinical applications, vertical measurements are difficult because of the length of the healing abutment and treatment position of patient. Further research is needed on how the ISQ and IST values change with implant length at different density. Further research is needed to analyze the reliability of the results based on expertise, such as the operator's experience in clinical practice.

## Conclusions

Within the limitation of this in vitro study,

1. In the artificial bone block, the primary stability of both devices was significantly higher in models with medium bone density, regardless of the implant length and observer.

2. At low bone density, primary stability improved with increasing implant length, whereas at medium density there was no significant difference in primary stability beyond 10 mm. This finding suggests that long implants can be an effective alternative to compensate for the primary stability of implants in patients with poor bone quality.

3. The results from both devices displayed similar trends regardless of bone density and implant length variations, with no differences between the devices.

4. Compared to Osstell® Beacon+, the simplicity of the measurement process makes Anycheck® easy and simple to use, regardless of the observer's expertise.

## Supporting information

**S1 Raw data.**
(XLSX)

**S2 Raw data.**
(XLSX)

**S3 Raw data.**
(XLSX)

## Author Contributions

**Conceptualization:** Jungwon Lee, Young-Jun Lim, Yeon-Wha Baek, Bum-Soon Lim.

**Data curation:** Jungwon Lee.

**Formal analysis:** Jin-Soo Ahn, Bongju Kim.

**Funding acquisition:** Young-Jun Lim.

**Investigation:** Jungwon Lee, Bongju Kim, Yeon-Wha Baek.

**Methodology:** Jungwon Lee.

**Supervision:** Yeon-Wha Baek, Bum-Soon Lim.

**Validation:** Jungwon Lee, Bongju Kim, Yeon-Wha Baek.

**Visualization:** Jungwon Lee, Jin-Soo Ahn.

**Writing – original draft:** Jungwon Lee, Young-Jun Lim.

**Writing – review & editing:** Young-Jun Lim, Yeon-Wha Baek, Bum-Soon Lim.

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
