## [Decision Letter · Decision Letter 0]

15 Oct 2023

PONE-D-23-25385Correlation of two different devices for the evaluation of primary implant stability depending on dental implant length and bone density: an in vitro studyPLOS ONE

Dear Dr. Baek,

Thank you for submitting your manuscript to PLOS ONE. After careful consideration, we feel that it has merit but does not fully meet PLOS ONE’s publication criteria as it currently stands. Therefore, we invite you to submit a revised version of the manuscript that addresses the points raised during the review process.

We look forward to receiving your revised manuscript.

Kind regards,

Sameh Attia, MS

Academic Editor

PLOS ONE

“This work was supported by the Korea Medical Device Development Fund grant funded by the Korea government (the Ministry of Science and ICT, the Ministry of Trade, Industry and Energy, the Ministry of Health & Welfare, the Ministry of Food and Drug Safety) (Project Number:1711138936, RS-2020-KD000291).”

5. "PLOS requires an ORCID iD for the corresponding author in Editorial Manager on papers submitted after December 6th, 2016. Please ensure that you have an ORCID iD and that it is validated in Editorial Manager. To do this, go to ‘Update my Information’ (in the upper left-hand corner of the main menu), and click on the Fetch/Validate link next to the ORCID field. This will take you to the ORCID site and allow you to create a new iD or authenticate a pre-existing iD in Editorial Manager. Please see the following video for instructions on linking an ORCID iD to your Editorial Manager account: https://www.youtube.com/watch?v=_xcclfuvtxQ

Reviewers' comments:

Reviewer's Responses to Questions

**Comments to the Author**

1. Is the manuscript technically sound, and do the data support the conclusions?

Reviewer #1: Yes

Reviewer #2: Partly

2. Has the statistical analysis been performed appropriately and rigorously? 

Reviewer #1: I Don't Know

Reviewer #2: I Don't Know

3. Have the authors made all data underlying the findings in their manuscript fully available?

Reviewer #1: Yes

Reviewer #2: Yes

4. Is the manuscript presented in an intelligible fashion and written in standard English?

Reviewer #1: Yes

Reviewer #2: No

5. Review Comments to the Author

Reviewer #1: The results are not unexpected. There are already innumerable studies dealing with the influence of implant size and bone density on primary stability. The correlation analysis between the measurement methods and the distinction between the experience of the examiner have been studied less frequently, but are also of minor interest clinically.

My comments:

Abstracts:

- Group 1 and 2 must be better explained.

Material & method

- More information regarding implant placement are necessary. Was placement done under axial load? Are implants self-drilling were used?

- Definition of an Expert or Non-Expert is missing.

- Information regarding sale size calculation, resulting normal distribution, as well as which post-hoc test was performed for the group comparisons are missing?

Discussion:

- Please explain why you used exactly these polyurethane blocks. They do not have a superficial layer to simulate the cortical bone. This makes the transfer of the obtained information into clinical reality even more difficult.

Reviewer #2: The manuscript ¨Correlation of two different devices for the evaluation of primary implant stability depending on dental implant length and bone density: an in vitro study¨, should be rewritten in several parts, mainly because the main objectives described are not in line with the conclusions.

The authors present results comparing experienced and non-experienced examiners, without referring to this as one of the objectives of the study, making the manuscript difficult to understand.

Regarding comparisons made on the dimensions of implants in relation to bone density, there are many studies previously published on this topic.

The discussion chapter is very poor.

6. PLOS authors have the option to publish the peer review history of their article (what does this mean?). If published, this will include your full peer review and any attached files.

Reviewer #1: No

Reviewer #2: No

---

## [Author Response · Author response to Decision Letter 0]

28 Nov 2023

Dear editor

Manuscript Number: PONE-D-23-25385

“Correlation of two different devices for the evaluation of primary implant stability depending on dental implant length and bone density: an in vitro study " 

We thank the reviewers for their comments, and we are grateful for the opportunity to provide further revisions to our paper. We changed our manuscript according to the reviewers’ comments and recommendations. We are trying to adequately address each of the points made by the reviewers. We would be very thankful if you could please reconsider a thoroughly revised manuscript. We highlighted the changes made in the manuscript by using a different color font (red): see correction marked form file, and explained details in this letter.

Response to Reviewer #1 Comments

1. Abstracts:

- Group 1 and 2 must be better explained.

Response: Thank you for your kind comments to our manuscript. Following sentence was revised in abstract as suggested by the reviewer.

Before: Group I and Group II had implants placed in an artificial bone model with a uniform density of 15 PCF (0.24 g/cm3) and 30 PCF (0.48 g/cm3), respectively.

After: In Group I, low-density bone was described using 15 PCF (0.24 g/cm3) polyurethane bone blocks, and in Group II, 30 PCF (0.48 g/cm3) polyurethane bone blocks were used to describe medium density bone.

2. Material & method

- More information regarding implant placement are necessary. Was placement done under axial load? Are implants self-drilling were used?

Response: Thank you for your valuable review of the manuscript. In this study, all implants used in the experiment were placed in bone blocks using a specially designed ‘implant placement & drilling machines’ (Hangil Technics, Gyeonggi, Korea), with self-tapping according to the protocol recommended by the manufacturer.

All contents of the “Surgical Procedure and Implant Placement” part of Materials & Methods have been carefully revised as suggested by the reviewer. 

Surgical Procedure and Implant Placement

Sixty implants were used, 30 in each group and they consisted of three different lengths (10 implants each): 7.3 mm, 10 mm, and 13 mm. All implants used in the study were inserted at a constant depth and vertical angle to the bone blocks using a specially designed implant placement & drilling machine (Hangil Technics, Gyeonggi, Korea) (Fig 2) with self-tapping for standardization.

The implant placement site was prepared using two drilling protocols according to the manufacturer’s instructions [12]. In Group I, a ∅ 2.2 mm initial drill and ∅ 3.0 mm taper drills were used [12], and in Group II, a ∅ 2.2 mm initial drill, final drills of ∅ 3.0 mm and ∅ 3.5 mm, and a ∅ 3.5 mm tap drill were used. Both groups used the Neo Master Kit (Neobiotech, Seoul, Korea) and were drilled at 1,200 rpm. The insertion torque was set around 18 Ncm for group I and 35-40 Ncm for group II. Each implant was placed 30 mm apart.

- Definition of an Expert or Non-Expert is missing.

Response: Thank you for your detailed review of our manuscript. I added the definitions of expert and non-expert in Materials and Methods. We marked all the corrections in red.

Before: The observer consisted of expert and non-expert.

After: The observer consisted of expert and non-expert. An expert refers to a dental hygienist with five years of experience proficient in using measuring device, while a non-expert denotes someone who has never used measuring device.

- Information regarding sale size calculation, resulting normal distribution, as well as which post-hoc test was performed for the group comparisons are missing?

Response: Thank you for your valuable comments to our manuscript. We used Scheffe and Duncan post hoc test for statistical analysis. We added above information in Statistical Analysis. 

Statistical Analysis

Before: One-way ANOVA was performed to verify that the ISQ and IST values of the three-implant length (7.3 mm, 10 mm, and 13 mm) reported a significant difference. 

After: One-way ANOVA was performed to verify that the ISQ and IST values of the three-implant length (7.3 mm, 10 mm, and 13 mm) reported a significant difference. Scheffe and Duncan were used for post hoc.

3. Discussion:

- Please explain why you used exactly these polyurethane blocks. They do not have a superficial layer to simulate the cortical bone. This makes the transfer of the obtained information into clinical reality even more difficult.

Response: Thank you for the valuable comments to our manuscript. Although RFA has been shown to be a useful tool for implant stability assessment, the ISQ values do not mirror the bone–implant contact at deeper parts, but rather at the marginal bone region. 

Previous studies [18-20], have suggested that a distinct layer of cortical bone on marginal bone plays a decisive role in the clinical value of RFA, whereas trabecular bone has a minor influence on the implant stability compared with the marginal bone density. 

18. Chávarri-Prado D, Brizuela-Velasco A, Diéguez-Pereira M, Pérez-Pevida E, Jiménez-Garrudo A, Viteri-Agustin I, Estrada-Marinez A, Montalbán-Vadillo O. Influence of cortical bone and implant design in the primary stability of dental implants measured by two different devices of resonance frequency analysis: An in vitro study. Journal of Clinical and Experimental Dentistry. 2020; 12: 242-248 

19. Chatvaratthana K, Thaworanunta S, Seriwatanachai D, Wongsirichat N. Correlation between the thickness of the crestal and buccolingual cortical bone at varying depths and implant stability quotients. PLos One. 2017; 12: 0190293-0190306 

20. Hsu JT, Fuh LJ, Tu MG, Li YF, Chen KT, Huang HL. The Effects of Cortical Bone Thickness and Trabecular Bone Strength on Noninvasive Measures of the Implant Primary Stability Using Synthetic Bone Models. Clincal Implant Dentistry. 2011; 251-261.

Therefore, we designed the experiment protocol to investigate the difference in ISQ values between uniform low-density bone and medium-density bone, without the influence of cortical bone. We sought to compare differences in implant stability due to pure variables of low and medium bone density, excluding other influences.

Following sentences were added in Discussion part.

Previous studies [18-20], have suggested that a distinct layer of cortical bone on marginal bone plays a decisive role in the clinical value of RFA, whereas trabecular bone has a minor influence on the implant stability compared with the marginal bone density. The cortical bone layer in biological samples, as well as the structure of the bone, affects implant stability [18]. Kanthanat Chatvaratthana et al. found that the ISQ value was highly correlated with cortical bone thickness [19]. Previous studies have used models with a cortical bone layer, so the results are not purely a function of variables such as the diameter or length of the implant. In fact, some studies have shown similar implant stability results when the bone density is different, but the cortical bone layer is the same thickness [20]. 

In this study, the experimental protocol was designed to completely exclude the influence of cortical bone and evaluate only the influence of implant length by using a uniform bone block without a cortical bone layer. By doing so, we sought to compare differences in implant stability due to pure variables of low and medium bone density, excluding other influences. To compare the primary stability of the implant, the cancellous bone blocks were 15 PCF (0.24 g/cm3), which depicts low-density bone, and 30 PCF (0.48 g/cm3), which depicts medium-density bone.

 

Response to Reviewer #2 Comments

1. The manuscript ¨Correlation of two different devices for the evaluation of primary implant stability depending on dental implant length and bone density: an in vitro study¨, should be rewritten in several parts, mainly because the main objectives described are not in line with the conclusions.

Response: Thank you for the valuable review of the manuscript. As suggested by the reviewer, we meticulously reviewed and revised the manuscript. All the modifications have been highlighted in red.

2. The authors present results comparing experienced and non-experienced examiners, without referring to this as one of the objectives of the study, making the manuscript difficult to understand.

Response: Thank you for your favorable review of the manuscript. we revised the manuscript by adding a comment about comparing the results of experts and non-experts to the topic. we marked all the corrections in red.

 Before: This study primarily aimed to compare the values obtained using two different devices for primary stability depending on the dental implant length and artificial bone density and to investigate the correlation of results from the two devices.

 After: This study primarily aimed to compare the values obtained by expert and non-expert using two different devices for primary stability according to dental implant length and artificial bone density and to investigate the correlation of results from the two devices

3. Regarding comparisons made on the dimensions of implants in relation to bone density, there are many studies previously published on this topic.

Response: Thank you for the valuable review of the manuscript. we agree with your opinion that there are many previous studies comparing implant stability in relation to bone density. 

However, our study is different from the existing studies mentioned below:

1. Completely excluding operator-related implant installation errors by using a specially designed implant placement & drilling machine (Hangil Technics, Gyeonggi, Korea) for standardization.

2. The experimental protocol was designed to completely exclude the influence of cortical bone and evaluate only the influence of implant length by using a uniform bone block without a cortical bone layer. By doing so, we sought to compare differences in implant stability due to pure variables of low and medium bone density, excluding other influences. 

3. Including usability test results between two devices through comparison of measurement values by experts and non-experts.

4. Comparing the indirect correlation between the ISQ value of Osstell® Beacon+ and the IST value of Anycheck® obtained under the same conditions, each applying different measurement principles. 

5. Assessing the clinical validity of the IST value measured by tapping the healing abutment attached to the implant fixture without additional installation of a smart peg was evaluated by comparing it with the ISQ value, a representative quantitative measure of implant stability.

We supplemented the discussion. we marked all the corrections in red.

4. The discussion chapter is very poor.

Response: Thank you for the valuable review of the manuscript. Following the reviewer's suggestions, we have comprehensively improved the content of the discussion, and all the revised content has been marked in red.

---

## [Decision Letter · Decision Letter 1]

28 Dec 2023

Correlation of two different devices for the evaluation of primary implant stability depending on dental implant length and bone density: an in vitro study

PONE-D-23-25385R1

Dear Dr. Baek,

We’re pleased to inform you that your manuscript has been judged scientifically suitable for publication and will be formally accepted for publication once it meets all outstanding technical requirements.

Kind regards,

Sameh Attia, MS

Academic Editor

PLOS ONE

Additional Editor Comments (optional):

Reviewers' comments:

Reviewer's Responses to Questions

**Comments to the Author**

1. If the authors have adequately addressed your comments raised in a previous round of review and you feel that this manuscript is now acceptable for publication, you may indicate that here to bypass the “Comments to the Author” section, enter your conflict of interest statement in the “Confidential to Editor” section, and submit your "Accept" recommendation.

Reviewer #1: All comments have been addressed

2. Is the manuscript technically sound, and do the data support the conclusions?

Reviewer #1: Yes

3. Has the statistical analysis been performed appropriately and rigorously? 

Reviewer #1: N/A

4. Have the authors made all data underlying the findings in their manuscript fully available?

Reviewer #1: Yes

5. Is the manuscript presented in an intelligible fashion and written in standard English?

Reviewer #1: Yes

6. Review Comments to the Author

Reviewer #1: The authors have answered and applied my comments to my satisfaction. I can recommend the acceptance.

7. PLOS authors have the option to publish the peer review history of their article (what does this mean?). If published, this will include your full peer review and any attached files.

Reviewer #1: No
